# Effect of Deep Cryogenic Treatment on Corrosion Behavior of AISI H13 Die Steel

**DOI:** 10.3390/ma14247863

**Published:** 2021-12-18

**Authors:** Tarang Shinde, Catalin Pruncu, Narendra B. Dhokey, Anca C. Parau, Alina Vladescu

**Affiliations:** 1Department of Mechanical Engineering, Finolex Academy of Management and Technology, Ratnagiri 415639, India; tarangshinde@gmail.com; 2Department of Mechanical Engineering, Imperial College London, Exhibition Rd., London SW7 2AZ, UK; 3Department of Metallurgy & Materials Science, College of Engineering, Shivajinagar, Pune 411005, India; nbdhokey@yahoo.co.in; 4Department of Advanced Surface Processing and Analysis by Vacuum Technologies (ReCAST), National Institute of Research and Development for Optoelectronics INOE 2000, 409 Atomistilor St., Magurele, 077125 Bucharest, Romania; anca.parau@inoe.ro; 5Physical Materials Science and Composite Materials Centre, Research School of Chemistry & Applied Biomedical Sciences, Tomsk Polytechnic University, Lenin’s Avenue 30, 634050 Tomsk, Russia

**Keywords:** corrosion, deep cryogenic treatment, AISI H13, Nyquist, open circuit potential, tool steels, corrosion rate

## Abstract

AISI H13 die steel specimens were subjected to heating at 1020 °C followed by oil quenching and double tempering at 520 °C. Subsequently, these specimens were subjected to deep cryogenic treatment at −185 °C in liquid nitrogen environment for 16 h and then subjected to soft tempering at 100 °C once the specimens attained room temperature. Thereafter, the specimens were subjected to scanning electron microscopy (SEM) analysis and electron backscatter diffraction (EBSD) analysis. The electrochemical corrosion activity was investigated in 3.5% NaCl at 23 ± 0.5 °C by evaluating the evolution of open circuit potential over time and potentiodynamic curves, and electrochemical impedance spectroscopy study was also carried out. The heat-treated specimens exhibited better resistance to corrosion through more electropositive values of open circuit potential. This could be attributed to lower grain boundary area in heat-treated specimens as compared to 16 h cryogenically treated specimen as higher grain boundary areas behave as an anode in an electrochemical cell, thereby enhancing the rate of corrosion. According to electrochemical tests, the cryogenically treated surface is more resistant to corrosion, followed by heated alloy. However, both surface modification treatments improved the corrosion behavior of the untreated alloy.

## 1. Introduction

The tool steels are widely used in hot working industries including forging and extrusion. According to the market study and analysis, the global market size of the tool steels was USD 4.29 billion in 2020, and as per the forecast, their value is expected to cross over USD 6395.95 by 2025 [1,2]. AISI H13 hot work die steel is one of the most important materials belonging to this category. In some of the applications, AISI H13 undergoes a phase change, austenitizing some region of the component when it is under stress and heat due to friction which could promote the spalling [3]; other reasons for failure could be oxidative or abrasive wear, fatigue cracking and chipping as experienced in some of the cases [4]. The corrosion damage of H13 die steel has also been reported in the case of applications involving the manufacturing of dies for aluminum cans [5]. Since these materials are subjected to heat and mechanical stresses, their corrosion behavior needs to be studied in order to investigate their response towards changing conditions, as well as to report any possible degradation in their properties during service.

The tool steels need to possess certain characteristics in order to withstand highly challenging conditions during mechanical working operations [6]. These characteristics include hardness, resistance to wear and fatigue properties. Even though the basic combinations present in tool steels do add up these characteristics, the need of the time is to further improve them. The deep cryogenic treatment has been recommended as an add-on process to improve certain mechanical properties, especially wear resistance, as indicated by many researchers in their studies on tool steels [7,8,9,10,11,12,13,14,15,16,17]. This treatment increases the density of precipitation of most of the carbides in tool steels and reduces average carbide particle size [18]. Some researchers have elaborated that the deep cryogenic treatment improves the surface finish of the materials through the refinement of martensite laths [19]. This method had been beneficial for materials such as AISI H13 die steel [20,21,22,23] in order to improve the wear resistance, hardness and surface characteristics, thereby enhancing the performance of the material.

The fine carbides precipitated after cryogenic treatment assist with the improvement of these properties substantially. The cryogenic treatment even had an impact on fatigue life, as reported by some researchers [24,25]. The fine carbides could act as microcrack arresters. They delay the propagation of these microcracks, thereby enhancing the fatigue life, which is quite beneficial for applications such as mechanical working. It has also been elaborated that the deep cryogenic treatment improved the corrosion resistance for some tool steels such as 1.2080 [26]. Corrosion behavior of hot work tool steels has been studied, and it was reported by researchers that methods such as pressure vapor deposition (PVD) and nitriding could be helpful in improving the corrosion resistance of these materials [27,28,29]. PVD coatings assist in localizing the attack on the material surface. A good adhesiveness is essential in retaining the coating layer and thereby preventing any further attack of the chemical or any surrounding agent which could result in formation of any corrosive byproduct.

One researcher experimented with the formation of a borided layer on the surface of AISI H13 steel [30]. The borided layers formed single phase or dual phase structures and their layers were compact and crack-free, which improved the corrosion resistance of the steel, especially in the case of steam turbine applications. The deep cryogenic treatment has been beneficial in improving the corrosion characteristics of structural steels. It could certainly become a potential method for those applications where corrosion plays a very important role deciding the lifespan of the component [31]. Most of the treatments have focused upon surface modifications in order to improve the corrosion resistance of tool steels.

In the last few decades, there was an interest in increasing the wear resistance of steels by applying deep cryogenic treatment due to homogenization and stabilization of internal microstructure. The modification of other steels such as D-2, M-2 and AISI H13 by deep cryogenic treatment can be found in the literature. The aim of the paper is to investigate the corrosion resistance of the AISI H13 die steel specimens by the deep cryogenic treatment at −185 °C in liquid nitrogen environment for 16 h and to see the effect of this treatment on its properties. Electron backscatter diffraction (EBSD) analysis, grain mapping tools and Nyquist plots have been used in the current research work in order to evaluate the performance of AISI H13 die steel specimens when subjected to a corrosive atmosphere.

The study also investigates the outcomes based on the experimentation to comment upon the best treatment for H13 steel in order to withstand the corrosive atmosphere. The results have been compared to those of the untreated and conventionally heat-treated specimens.

## 2. Materials and Methods

### 2.1. Surface Modification Process

AISI H13 material was procured from Rajasthan Steels, Pune (India), to fabricate the specimens for research work. The chemical composition involves C—0.39%, Mn—0.30%, Si—0.40% Cr—5.40%, Mo—1.4%, V—1.00%, balance Fe, by weight. A set of AISI H13 specimens (diameter 8 mm and length 15 mm) was hardened at 1020 °C with a step-by-step heating (first heating to 500 °C at a rate of 7 °C/min and holding for 20 min and thereafter heating to 860 °C and soaking for 20 min) to prevent cracking due to thermal stresses. These specimens were soaked for 20 min at 1020 °C followed by oil quenching. The quenched specimens were double tempered at 520 °C for 2 h. This treatment was named conventional heat treatment for AISI H13 steel and the specimens were designated as “heated” specimens. In a cryo-chamber (Sanmar, Mumbai, India) supplied with liquid nitrogen at a cooling rate of 3 °C/min, the other set of specimens was subjected to conventional heat treatment (step-by-step heating to 1020 °C, oil quenching and double tempering at 520 °C for 2 h) and cryogenic treatment at −185 °C at a cryo soaking period of 16 h [21]. At the end of 16 h of cryo soaking period, these specimens were transferred to an insulated box (Cryobox, Sigma-Aldrich, Irvine, UK) until they attained room temperature, and thereafter the specimens were soft-tempered for 1 h at 100 °C to eliminate the cold stresses produced during the cryogenic treatment. Specimens cryogenically treated for 16 h were referred to as “cryo” specimens.

### 2.2. Characterization Methods of Developed Surfaces

All specimens were examined by scanning electron microscopy (SEM) (Tabletop Microscope TM3030, Hitachi, Tokyo, Japan) after mechanical polishing which involved grinding down using SiC papers (homemade), sequentially from 320 to 4000 grits. Subsequently, the ground specimens were active oxide polishing (OPS) polished for 30 min using a suspension diluted with H_2_O with a ratio of 1:5 to a mirror finish.

The specimens were further characterized for grain orientation and texture direction using a Hitachi 3400 SEM-based Bruker e-flash electron backscatter diffraction (EBSD) detector (Cheshire, UK) at 20 kV acceleration voltage, 10 mA current density and 1 mm step size. The images were postprocessed using MTEX software (free Matlab toolbox, see details in https://mtex-toolbox.github.io/, accessed on 3 December 2021).

The electrochemical behavior was measured using a VERSASTAT potentiostat/galvanostat (Princeton Applied Research, Princeton, NJ, USA). The tests were performed in 3.5% NaCl solution (pH = 7.4) at 23 ± 0.5 °C. Each sample was placed in a Teflon sample holder (Princeton Applied Research, Princeton, USA) with exactly 1 cm^2^ exposed to the corrosive media. A platinum electrode (XM140, Radiometer Analytical, Loveland, CO, USA) was used as the counter electrode and saturated calomel (XR110, Radiometer Analytical, Loveland, CO, USA) was used as the reference electrode. All measurements were achieved at a scanning rate of 1 mV/s. The open circuit potential (E_OC_) was monitored for 1 h, starting right after the immersion of the sample in the 3.5% NaCl solution, and the potentiodynamic curves were recorded at −1 to +2 V vs. E_OC_. The tests were performed according to the standard ISO 16151:2018. Electrochemical behavior of the investigated specimens was also examined by electrochemical impedance spectroscopy (EIS). Impedance measurements were performed at open circuit potential with constant perturbing AC signal amplitude of 10 mV over a frequency range extending from 0.1 to 10^4^ Hz. Analysis of the spectra was performed by equivalent circuit fitting using Zview software (Princeton Applied Research, Princeton, USA). After the corrosion tests, each surface was analysed by SEM and profilometry (Dektak 150, Bruker, Billerica, MA, USA) in order to evaluate the morphology and roughness.

## 3. Results

### 3.1. Microstructural Results

The microstructural details of polycrystalline materials can be investigated with the help of electron backscatter diffraction (EBSD) analysis [32]. One of the reasons why grain boundaries must be analyzed is that the performance and integrity of the material depend upon the grain boundary networks present inside the materials [33]. The EBSD feature has made it possible to analyze more than 10^4^ boundaries in order to investigate their role in properties such as corrosion resistance [34,35]. The grain size has a significant impact on the mechanical and corrosion properties of an alloy. The EBSD phase maps could be useful in identification and distribution of different phases in the alloys. Figure 1 shows the phase maps for untreated, heated and cryo-treated specimens depicting the significant precipitation of different phases in different cases. Figure 2 reports the results obtained through EBSD analysis clarifying that the heat treatment (Figure 2b) and cryogenic treatment (Figure 2c) bring about recrystallized structure in the material. A low angle grain boundary is typically the boundary between two crystal grains with a misorientation of less than 15° [36,37].

Figure 3 elaborates the low angle grain boundary (LAGB) patterns for untreated, heated and cryo specimens which indicate that the as-received specimen has a limited number of LAGBs (noted by green) while heat-treated and cryo specimens show more LAGBs which could be attributed to the recrystallized structure in both specimens.

The deep cryogenic treatment of H13 steel resulted in a fine-grained structure. As geometrically necessary dislocation (GND) density measurement by EBSD has become a very popular tool in microstructural analysis [38], Figure 4 presents the geometrically necessary dislocation evolution of untreated, heated and cryo-treated specimens. It is observed from the GND maps that the dislocation density trend is untreated < heated < cryo; the highest dislocation density is found for cryo specimens (10^14^–10^15^), which could be attributed to increased plastic strain, resulting in accumulation of dislocations at the grain boundary areas for cryo specimens of H13 steel.

Figure 5 represents the grain size variation for untreated, conventionally heat-treated (heated) and 16 h cryogenically treated (cryo) specimens. The graph illustrates that the grain size is minimal for 16 h cryogenically treated H13 steel for both conditions (before and after corrosion). The cryogenic treatment assists in the precipitation of fine carbides along the grain boundaries which inhibits the grain growth, resulting in a fine-grained structure [39]. Minimum grain size depicts the maximum grain boundary area inside the material. It has been shown by some authors that the grain refinement leads to increased susceptibility to corrosion [40,41,42], but the passivity of the film present on the surface plays a vital role in deciding the response of the material to corrosion. When untreated specimens are compared with conventionally treated (heated) ones, there is a reduction in grain size and, as stated by some authors, the corrosion resistance would be greater if the passivity of the film is maintained even for a fine-grained material [43,44,45].

### 3.2. Electrochemical Results

The electrochemical corrosion activity was investigated in 3.5% NaCl at 23 ± 0.5 °C by evaluating the evolution of open circuit potential (OCP) over time (Figure 5) and potentiodynamic curves (Figure 6). The open circuit potential (OCP) is a parameter that is related to the protective ability of the passive film. During 1 h of immersion, the OCP value slightly changed, indicating that steady-state conditions were not reached (Figure 5). Based on the results presented in Figure 6, one may see that the “cryo” specimen has an electropositive value of E_oc_, indicating a good resistance to NaCl attack.

Figure 7 represents the polarization curves of the investigated surfaces. Based on Tafel extrapolation [46], the main corrosion parameters were extracted from Figure 6 and presented in Table 1, as frequently reported [47]. Heated specimens exhibited a more electropositive value of the corrosion potential (E_i_ = 0) compared to the uncoated and cryo specimens, indicating that the corrosive solution had less influence on their surfaces. The cryo specimen had a more electronegative corrosion potential, showing poor corrosion resistance to NaCl attack. Lower corrosion current density (i_corr_) and higher polarization resistance (R_p_) values were observed for heated specimens, showing good corrosion resistance. Taking into account the electrochemical parameters, it can be said that heated specimens exhibited the best corrosion resistance to 3.5% NaCl, followed by the untreated and the cryo specimens.

The surface porosity (P) was estimated based on Elsner’s empirical equation [48,49] taking into account the values of polarization resistance (R_p_) before and after applied treatments. Moreover, the protective efficiency (P_e_) was also calculated based on the formula reported in [49], considering the ion corrosion densities of the untreated substrate and treated substrates. The corrosion rate (CR) has been estimated according to ASTM G102-89 standard (reapproved 2015) [50], using the following equation:(1)$$\mathrm{CR}=K_{i}\;\frac{i_{\mathrm{corr}}}{\rho }\;\mathrm{EW}$$
where CR = corrosion rate, K_i_ = 3.27 × 10^−3^, ρ = materials density, i_corr_ = corrosion current density and EW = equivalent weight.

Taking into account the values of porosity, one may see that the alloy exhibited low porosity after heating, compared with the cryogenically treated alloy. Moreover, the protective efficiency in response to NaCl attack is higher in the case of the heated alloy when compared to the cryo alloy. The corrosion rate can be estimated in the following order: untreated > cryo > heated alloy. The high corrosion rate of cryogenically treated alloy can be attributed to the high porosity of the surface. However, it can be summarized that the best corrosion resistance was found for the heat-treated alloy and cryogenically treated alloy.

Electrochemical impedance spectroscopy (EIS) was used to investigate the electrochemical behavior of the investigated systems, highlighting the surface properties. For this purpose, the applied amplitude of the perturbation signal was 10 mV RMS vs. Eoc in a frequency range of 0.1–10^4^ Hz. The impedance data were displayed as a Nyquist plot (Figure 8). According to this plot, one may observe that the specimens exhibit a better protection with increasing order from left to right as cryo < untreated < heated.

### 3.3. Morphology before and after Electrochemical Tests

The morphology before and after electrochemical tests was evaluated by SEM analysis, and the results are presented in Figure 9. Before electrochemical tests, all surfaces appeared to be without defects or cracks. After corrosion tests, each surface was affected differently. The surface of the untreated alloy was destroyed on all corroded areas. At the magnification of 5000×, some precipitation of oxides of Al, Cr and Fe can be seen, as can the NaCl precipitates from the corrosive solution. The heated alloy is more resistant to corrosion, having a localized corrosion in some parts of the corroded region. On this surface, some oxides of Al, Cr and Fe can be seen, having a high affinity to oxygen. Some holes can also be observed, indicating that the corrosive solution penetrated more deeply. In the case of cryogenically treated alloy, many cracks appeared on the surface after the electrochemical tests. One possibility is that these cracks could be attributed to the destruction of the oxides. The heated AISI H13 steel displays better corrosion resistance compared to untreated AISI H13, and their corrosion resistance is comparable to that of the cryogenically treated alloy. In order to explain more about the corrosion process, the investigation of the roughness of the specimens was performed before and after corrosion tests.

### 3.4. Roughness before and after Electrochemical Tests

The roughness was evaluated on a length of 10 mm by a surface profilometer (Dektak 150, Bruker, Billerica, MA, USA) equipped with a 2.5 µm diameter stylus. For evaluation of roughness, two main parameters were calculated: arithmetic average deviation from the mean line (R_a_) and skewness (S_kew_). In Figure 10, the histograms of both calculated roughness parameters captured before and after electrochemical tests are presented. According to the R_a_ parameter, one may note that the roughness of all investigated surfaces was significantly increased after corrosion tests, indicating that all surfaces were affected by the corrosive solution. The heated and cryo alloys exhibited almost similar values of R_a_, indicating that both surfaces exhibited comparable corrosion behavior. More relevant is the skewness parameter. A negative value of S_kew_ implies a surface formed of many valleys, while a positive value suggests a surface formed of mainly peaks and asperities. A S_kew_ with a value close to 0 suggests a flat surface. Taking into consideration these states, one may see that the untreated surface has an almost flat surface before corrosion and a surface with many peaks after corrosion. This finding demonstrated that the corrosive solution digs into the material and forms many pits. The heated alloy has a negative S_kew_ before corrosion tests, meaning that the surface has many valleys, which are transformed to high peaks after corrosion tests, indicating that the surface is significantly affected by the corrosive solution. The same phenomenon was also found in the case of the cryogenically treated surface. Note that the differences found for the cryo surface are small compared to those found for the heated surface, indicating that the cryo surface is more resistant to corrosion in 3.5% NaCl.

## 4. Discussion

Based on the results obtained through corrosion tests, it can be said that the grain boundaries play a very important role in deciding the properties of materials under different service conditions for all types of treatments. High geometrically necessary dislocation (GND) density has been noted for 16 h cryogenically treated AISI H13 specimens (cryo specimens), which could be attributed to the increased plastic strain or strain hardening resulting in the accumulation of dislocations at the grain boundary areas. A reduction in grain size has been observed in the case of 16 h cryogenically treated specimens (cryo specimens), which could be attributed to the precipitation of fine carbides at the grain boundaries inhibiting the grain growth [12]. The highest values of open circuit potential (OCP) for AISI H13 cryo specimens indicate that these specimens are good in resisting the attack of NaCl as compared to untreated and heated specimens. The heated specimens have shown good results in the case of polarization curves with Tafel’s extrapolation and estimation of surface porosity as compared to heated and cryo specimens. This indicates that the heated specimens form a protective and passive layer on the surface of the specimen. The Nyquist plots are also in agreement with the results for heated specimens. The SEM plots indicate a broken film in the case of cryo specimens, which could have resulted in increased corrosion; however, the reasons for the breakage of the film need to be determined.

Surface roughness evaluations are essential in analysing the response of materials to corrosive environments. There was a significant increase in the roughness for all types of specimens as an effect of the corrosive atmosphere. The skewness parameters evaluated for untreated, heated and cryo specimens indicate that the cryo specimens have better corrosion resistance as compared to the other two types, which could be attributed to the less significant impact of the corrosive atmosphere on cryo specimens.

One of the important aspects that need to be discussed is the influence of mechanical properties on the corrosion behavior of the materials. In this case, if the hardness is compared amongst the three categories (i.e., untreated, heated and cryo-treated), it can be seen that the hardness increases as follows: untreated (42 HRC) < heated (50 HRC) < cryo (56 HRC). The increase in hardness could be the result of the conversion of retained austenite into martensite after cryogenic treatment and the precipitation of fine carbides in the matrix of tempered martensite [12,21]. The grain refinement is also one of the reasons for the improvement in hardness. Based on these results for hardness, it can be stated that increased martensitic content indicates better corrosion resistance as in the case of cryo specimens [51].

In summary, heated and cryo specimens of AISI H13 exhibit better resistance to corrosion as compared to untreated material, and deep cryogenic treatment is useful in improving the corrosion resistance.

## 5. Conclusions

The current research work was aimed at investigating the corrosion behavior of cryogenically treated AISI H13 die steel. The following points summarize the major outcomes:

The deep cryogenic treatment is beneficial in improving the corrosion resistance of AISI H13 material and could be useful in applications involving high mechanical stress and a corrosive environment.The grain boundaries decide the corrosion characteristics of the material; increased geometrically necessary dislocations indicate higher plastic straining. There is a reduction in grain size for AISI H13 specimens subjected to 16 h deep cryogenic treatment, which is attributed to the precipitation of fine carbides at the grain boundary areas.Parameters such as higher open circuit potential and skewness indicate that H13 specimens subjected to deep cryogenic treatment have a better response to a corrosive environment, whereas the surface porosity measurements and Nyquist plots confirm the superior response of heated specimens to a corrosive atmosphere.

Overall, it can be concluded that deep cryogenic treatment is useful in improving the corrosion properties of H13 die steel. Some further investigations of material surfaces exposed to a high-temperature corrosive environment could also reveal some important outcomes for H13 steel.

## Figures and Tables

**Figure 1 materials-14-07863-f001:**
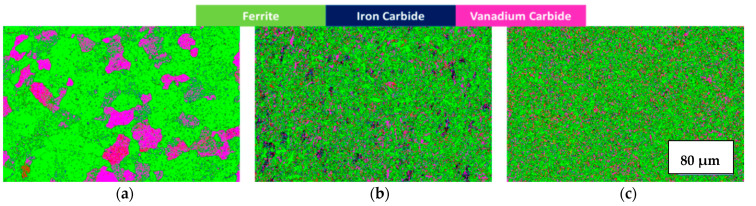
Maps depicting the main phase of studied material: (**a**) untreated, (**b**) heated and (**c**) cryo specimens.

**Figure 2 materials-14-07863-f002:**
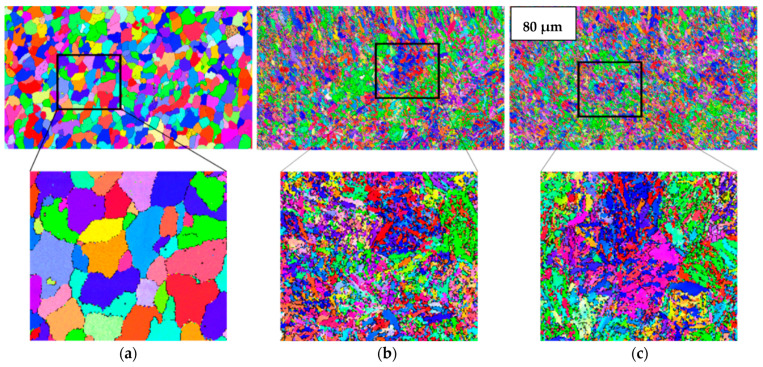
Inverse pole figure map obtained from EBSD analysis for (**a**) untreated, (**b**) heated and (**c**) cryo specimens.

**Figure 3 materials-14-07863-f003:**
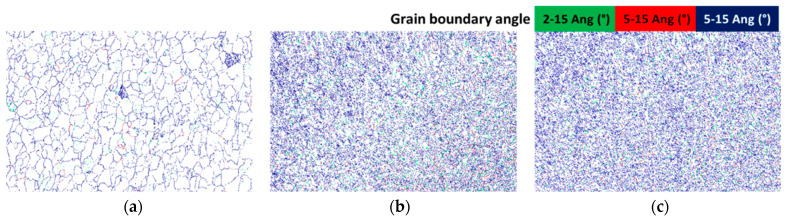
LAGBs for (**a**) untreated, (**b**) heated and (**c**) cryo specimens.

**Figure 4 materials-14-07863-f004:**
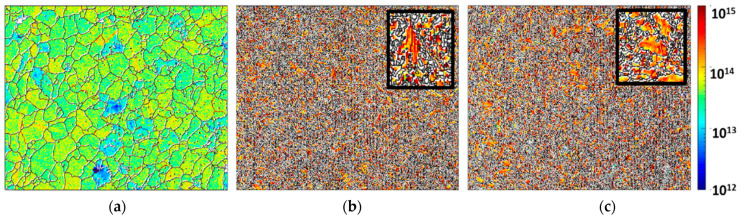
Geometrically necessary dislocation evolution of (**a**) untreated, (**b**) heated and (**c**) cryo-treated specimens.

**Figure 5 materials-14-07863-f005:**
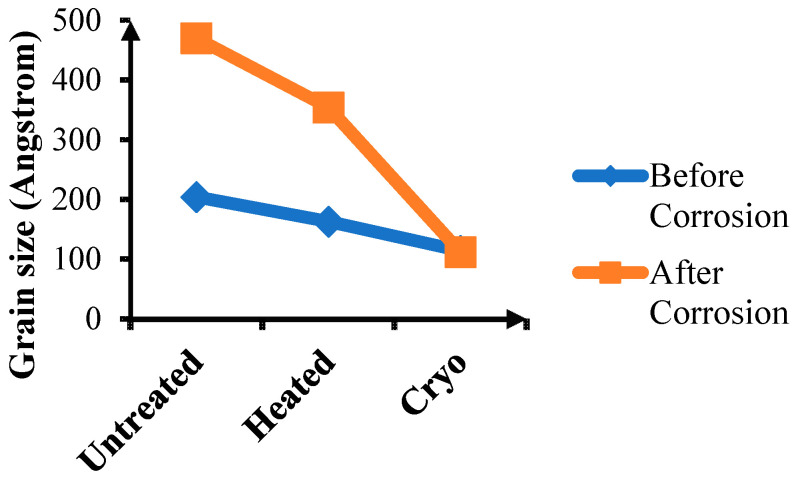
Grain size variation for untreated, conventionally heat-treated and 16 h cryogenically treated specimens before and after corrosion.

**Figure 6 materials-14-07863-f006:**
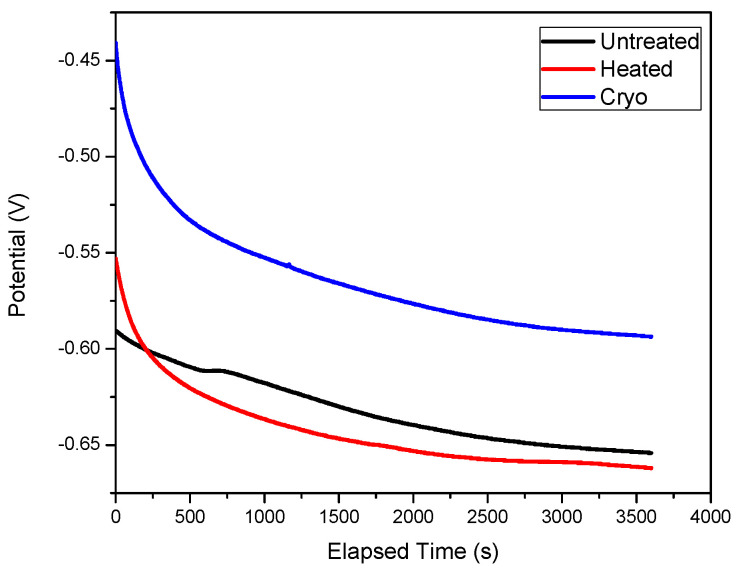
Evolution of the open circuit potential over time.

**Figure 7 materials-14-07863-f007:**
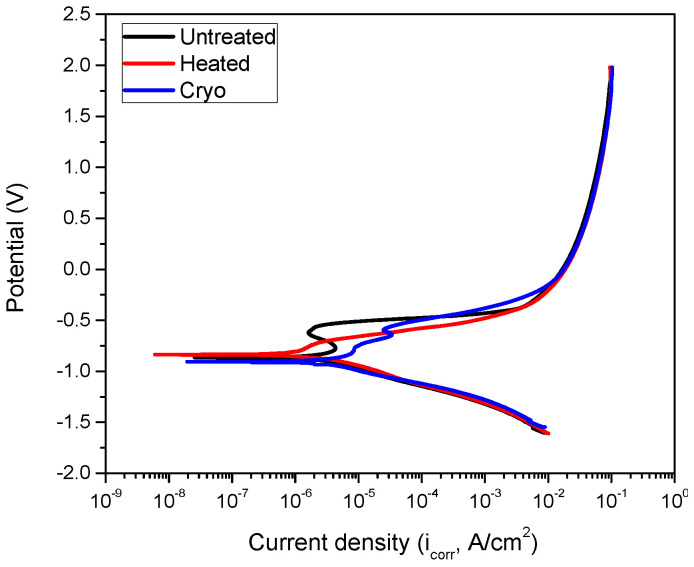
Potentiodynamic curves of investigated surfaces.

**Figure 8 materials-14-07863-f008:**
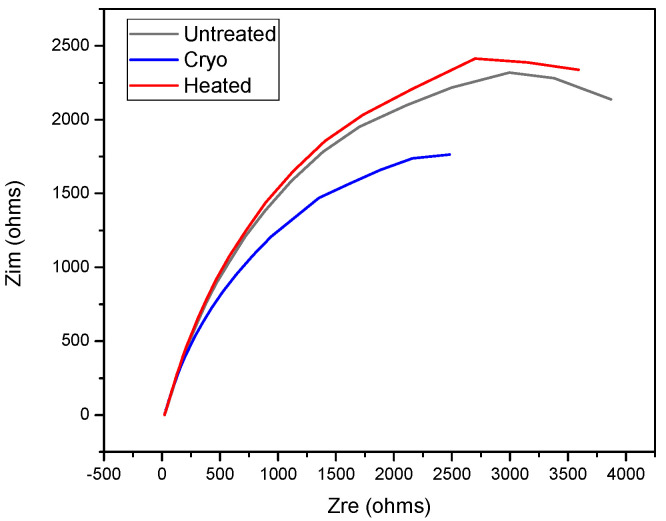
Nyquist plot for the investigated specimens.

**Figure 9 materials-14-07863-f009:**
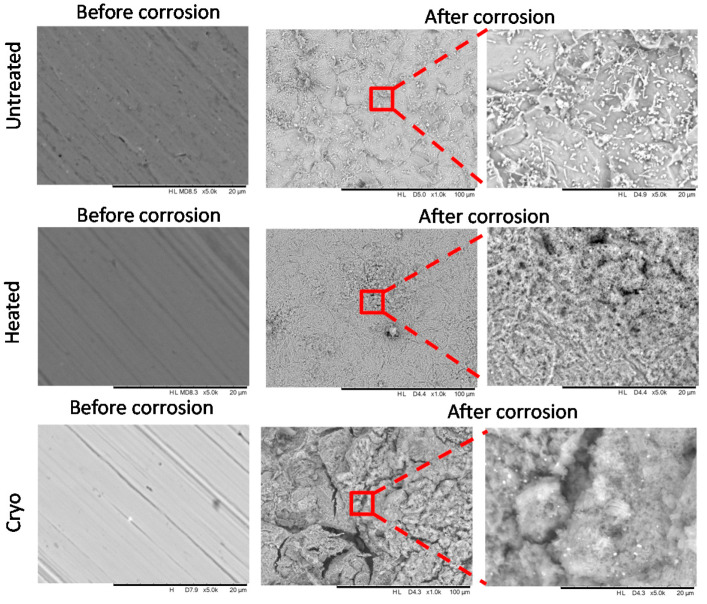
SEM images of the investigated specimens before and after corrosion.

**Figure 10 materials-14-07863-f010:**
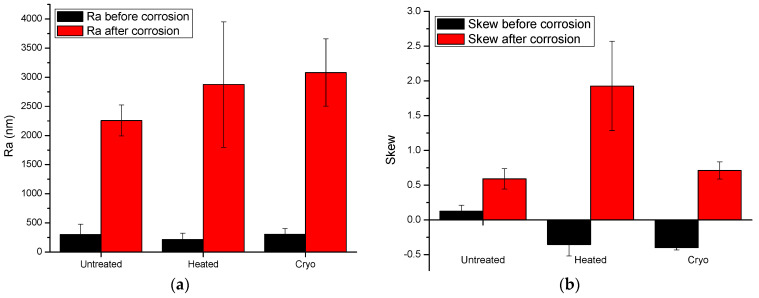
Roughness parameters: (**a**) R_a_—arithmetic average deviation from the mean line; (**b**) S_kew_—skewness of the investigated specimens before and after corrosion resistance.

**Table 1 materials-14-07863-t001:** Main corrosion parameters of untreated, heated and cryo specimens. E_i_ = 0: corrosion potential; i_corr_: corrosion current density; R_p_: polarization resistance; P: porosity; P_e_: protective efficiency; CR: corrosion rate.

Sample	E_i_ = 0 (mV)	i_corr_ (µA/cm^2^)	R_p_ (kΩ)	P	P_e_ (%)	CR (µm/year)
Untreated	−865	4.387	11.644	-	-	0.051
Heated	−835	1.303	20.917	0.532	70.3	0.015
Cryo	−906	2.005	9.130	1.199	54.3	0.023

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
