# Peer review of "Effect of Deep Cryogenic Treatment on Corrosion Behavior of AISI H13 Die Steel"

_materials, 2021, doi:10.3390/ma14247863_

Round 1

Reviewer 1 Report

The manuscript aims at improving the corrosion resistance of the AISI H13 die steel specimens using a deep cryogenic treatment. The specimen’s performances have been well investigated (electrochemical corrosion, morphology and roughness). The used procedures (surface modification process and the characterization methods) and the analysis have been deeply described. The authors demonstrate that heated and cryo specimens show a better corrosion resistance.

The manuscript is clear and the analysis well structured. The state of the art is also well described. The methodologies used are reported in detail and this allows the reproducibility of the results.

Few comments about figures to review:

  • Please provide higher magnification for Figures 1, 2, 3, 4 (Maps and text as well. For instance, text in the red box in Fig.3 is not readable).
  • “Figure 4. Grain size variation for untreated, conventionally heat treated and 16 hours cryogenically treated specimens before and after corrosion.” has to be renamed Figure 5 and the quality of the graph improved.
  • As a consequence, “Figure 5. Evolution of the open circuit potential in time.” should be Figure 6. Same comment for “Figure 6. Potentiodynamic curves of investigated surfaces.”, “Figure 7. Nyquist for the investigated specimens.” Please check all figures numeration and their citation in the text.
  • Please use the same font in the figures when adding (a), (b)…. For instance, the font is bigger in the current Fig. 9 than Fig. 1.

Author Response

Dear Dr. Su Xinran,

We would like to show our great gratitude to the reviewer for the useful comments and constructive suggestions on our manuscript, which do help us significantly improve the quality of the current paper. All the review comments are appreciated. We do benefit a lot from the suggestions/ comments to improve the quality of our manuscript. We have revised our manuscript accordingly. The revision of the paper was highlighted by the blue coloured font. Detailed and point-to-point response to the reviewer’s comments is summarized below.

Here, we re-submit a new version of our manuscript which has been checked and modified after our careful referring to the reviewers’ comments. Meanwhile, efforts were also made to improve the English of the paper. We hope all of these changes will make this manuscript accepted by reviewer. Thank you for your kind consideration.

Best regards,

Dr. Eng. Alina Vladescu

Reviewer 1

The manuscript aims at improving the corrosion resistance of the AISI H13 die steel specimens using a deep cryogenic treatment. The specimen’s performances have been well investigated (electrochemical corrosion, morphology and roughness). The used procedures (surface modification process and the characterization methods) and the analysis have been deeply described. The authors demonstrate that heated and cryo specimens show a better corrosion resistance.

The manuscript is clear and the analysis well structured. The state of the art is also well described. The methodologies used are reported in detail and this allows the reproducibility of the results.

Thank you very much for your useful comments and constructive suggestions on our manuscript, which do help us significantly improve the quality of the current paper. All the review comments were appreciated. We have revised our manuscript accordingly. Our answers are highlighted by the blue coloured font.

Few comments about figures to review:

  • Please provide higher magnification for Figures 1, 2, 3, 4 (Maps and text as well. For instance, text in the red box in Fig.3 is not readable).

Thank you for this suggestion. Figures 1,2,3,4 and the related text have been modified with proper magnification in the modified manuscript.

  • “Figure 4. Grain size variation for untreated, conventionally heat treated and 16 hours cryogenically treated specimens before and after corrosion.” has to be renamed Figure 5 and the quality of the graph improved.

Thank you for this suggestion. The quality of the graph has been improved and it is replaced with a high-quality image. The figure number has also been changed to Fig. 5. We are sorry for this mistake.  

  • As a consequence, “Figure 5. Evolution of the open circuit potential in time.” should be Figure 6. Same comment for “Figure 6. Potentiodynamic curves of investigated surfaces.”, “Figure 7. Nyquist for the investigated specimens.” Please check all figures numeration and their citation in the text.

We are sorry for this mistake. All figure numerations as well as their citations have been modified properly.

  • Please use the same font in the figures when adding (a), (b)…. For instance, the font is bigger in the current Fig. 9 than Fig. 1.

The font for (a) and (b) in all figures has been replaced.

Reviewer 2 Report

The article can be considered for publication after the following concerns are addressed.

  1. The overall command of English is ok, but specifically, the language used in the introduction needs to be polished. Some sentences look abrupt without proper transition between them. In addition, do consider separating the introduction into paragraphs.
  2. Several acronyms (e.g. SCE, RE, CE …) were defined and only used once as when they were first defined. It would not be necessary to define such acronyms.
  3. Line 195, I think authors meant Fig. 4.
  4. For Fig. 5, any insights on the cross-over, or the varying gradients, for the heated and untreated lines?
  5. Line 338: Why does martensitic transformation take place upon cyrogenic treatment? Citations are needed to support this.
  6. Line 341: Are there any proofs that the corrosion resistance of martensite is better than austenite? The reviewer thinks that it is the other way round actually.

Author Response

Dear Dr. Su Xinran,

We would like to show our great gratitude to the reviewer for the useful comments and constructive suggestions on our manuscript, which do help us significantly improve the quality of the current paper. All the review comments are appreciated. We do benefit a lot from the suggestions/ comments to improve the quality of our manuscript. We have revised our manuscript accordingly. The revision of the paper was highlighted by the blue coloured font. Detailed and point-to-point response to the reviewer’s comments is summarized below.

Here, we re-submit a new version of our manuscript which has been checked and modified after our careful referring to the reviewers’ comments. Meanwhile, efforts were also made to improve the English of the paper. We hope all of these changes will make this manuscript accepted by reviewer. Thank you for your kind consideration.

Best regards,

Dr. Eng. Alina Vladescu

Reviewer 2

The article can be considered for publication after the following concerns are addressed.

Thank you very much for your useful comments and constructive suggestions on our manuscript, which do help us significantly improve the quality of the current paper. All the review comments were appreciated. We have revised our manuscript accordingly. Our answers are highlighted by the blue coloured font.

  1. The overall command of English is ok, but specifically, the language used in the introduction needs to be polished. Some sentences look abrupt without proper transition between them. In addition, do consider separating the introduction into paragraphs.

The introduction part has been modified and necessary constructional changes have been done in the modified manuscript.

  1. Several acronyms (e.g. SCE, RE, CE …) were defined and only used once as when they were first defined. It would not be necessary to define such acronyms.

Thank for this suggestion. The acronyms have been removed from the modified manuscript since the related terms were used only once.

  1. Line 195, I think authors meant Fig. 4.

We are sorry for this mistake. The figure numbers have been changed and respective modifications have been done in the modified manuscript.

  1. For Fig. 5, any insights on the cross-over, or the varying gradients, for the heated and untreated lines?

The change in the gradient could be attributed to the heat treatment carried out on heated specimens hardening and double tempering. 

  1. Line 338: Why does martensitic transformation take place upon cyrogenic treatment? Citations are needed to support this.

Normally, after quenching, the austenite is transformed to martensite but the entire conversion of austenite to martensite is not taking place as in case of alloy steels like AISI H13, the martensite formation (Mf) temperature is below 0⁰C and during quenching, material attains room temperature. Some amount of retained austenite is present. During cryogenic treatment, the material is subjected to -185⁰C temperature. This retained austenite is converted to martensite since the material has crossed Mf temperature. The related citations have been mentioned in the modified manuscript (Reference No .12 and 21).

  1. Line 341: Are there any proofs that the corrosion resistance of martensite is better than austenite? The reviewer thinks that it is the other way round actually.

The authors have not commented upon the corrosion resistance of martensitic structure is better than austenitic. The corrosion resistance is better for martensitic structure (Reference 51)
